# Potential Protective Role of Galectin-3 in Patients with Gonarthrosis and Diabetes Mellitus: A Cross-Sectional Study

**DOI:** 10.3390/ijerph191811480

**Published:** 2022-09-13

**Authors:** Jelena Milosevic, Milena Jurisevic, Vesna Grbovic, Ivan Jovanovic, Nevena Gajovic, Aleksandra Jurisic-Skevin

**Affiliations:** 1Department of Physical Medicine and Rehabilitation, Faculty of Medical Sciences, University of Kragujevac, 34000 Kragujevac, Serbia; 2Clinic of Orthopedics and Traumatology, University Cinical Center of Kragujevac, 34000 Kragujevac, Serbia; 3Department of Clinical Pharmacy, Faculty of Medical Sciences, University of Kragujevac, 34000 Kragujevac, Serbia; 4Department of Physical Medicine and Rehabilitation, University Clinical Center of Kragujevac, 34000 Kragujevac, Serbia; 5Center for Molecular Medicine and Stem Cell Research, Faculty of Medical Sciences, University of Kragujevac, 34000 Kragujevac, Serbia

**Keywords:** gonarthrosis, diabetes mellitus, galectin-3, cytokines

## Abstract

Background: Gonarthrosis and diabetes mellitus are two diseases that are increasingly being linked. The aim of this study was to quantify serum levels of Gal-3, pro- and anti-inflammatory cytokines (including their ratios and correlations), and participant’s condition (pain, stiffness, functional limitations) in gonarthrosis patients with and without diabetes mellitus. Methods: A between-subject, cross-sectional experimental design was adopted. Serum levels of TNF-α, IL-6, IL-12, IL-23, IFN-γ, IL-17, IL-10, Gal-3, and WOMAC score were measured. Results: Gonarthrosis patients with diabetes mellitus had significantly (*p* < 0.05) lower levels of TNF-α, IL-6, IL-12, IL-17, IFN-γ, and Gal-3 compared to gonarthrosis patients without diabetes mellitus. On the other hand, IL-10/TNF-α, IL-10/IL-6, IL-10/IL-12, Gal-3/TNF-α, Gal-3/IL-6, and Gal-3/IL-12 (*p* = 0.001) were significantly higher (*p* < 0.05) in gonarthrosis with diabetes mellitus. Moderate–large correlation (*p* < 0.05) was detected between the serum values of Gal-3 and pro- and anti-inflammatory cytokines, including IL-12 (r = 0.575), IL-10 (r = 0.535), TNF-α (r = 0.306), and IL-23 (r = 0.323). WOMAC index was significantly lower (*p* < 0.05) in gonarthrosis patients without diabetes mellitus compared to gonarthrosis patients with diabetes mellitus. Conclusions: Correlation between Gal-3 and proinflammatory cytokines and its dominance over proinflammatory cytokines implicate the potential role of Gal-3 in preventing cartilage destruction.

## 1. Introduction

Gonarthrosis of the knee is a degenerative rheumatic disease characterized by the decomposition of cartilage, formation of bone growths on the edge of the joint, sclerosis of the subchondral bone, and changes in the joint capsule [1,2]. The most significant clinical symptoms of gonarthrosis are pain, stiffness, and decreased functional capacity of the knee [3,4]. Numerous factors can influence the development of gonarthrosis, including obesity, age, sex, genetics, muscle function, and occupation [3,4,5]. Gonarthrosis is associated with significant morbidity and costs, and it is the leading cause of disability [6,7]. Although gonarthrosis has been thought to be a noninflammatory degenerative disease, at present, a growing body of evidence points to the relationship between inflammation and the development of gonarthrosis [8,9].

Further, epidemiological studies showed an increasing number of patients with gonarthrosis who suffer from DM [10]. This can be explained by the fact that both diseases share a large number of the same risk factors for their occurrence, as well as the fact that both diseases have a very high prevalence [11,12,13]. Diabetes mellitus is one of the most common chronic diseases. It is characterized by chronic hyperglycemia with disturbance of carbohydrate, lipid, and protein metabolism. High glucose levels can lead to damage to connective tissue and the musculoskeletal system, by primarily affecting impaired cell function [14,15]. Two pathogenic pathways can explain the effect of DM on occurrence of gonarthrosis. Chondrocytes that are exposed to a high level of glucoses are unable to regulate the transport of glucoses, which leads to accumulation of glucoses and production of reactive oxidative species (ROS) [16]. Increased production of ROS stimulates development of oxidative stress and increases the expression of proinflammatory mediators that are responsible for changes in the structure of joint cartilage [3,16]. Insulin resistance locally or systemically leads to the occurrence of a low degree of infection, which can lead to remodeling of the subchondral bone and the occurrence of gonarthrosis [3,13,17].

Gal-3 is widely distributed in many cells in the human body, including immune cells, epithelial cells, endothelial cells, and sensory neurons [18]. Depending on the location in the cell, Gal-3 performs different functions. Gal-3, located in the cytoplasm, is important for cell survival, while presented in the nucleus it regulates gene transcription, and extracellularly located Gal-3 modulates cell-cell interactions [18,19]. Gal-3 has a pivotal role in numerous biological activities, such as powerful regulation of cell adhesion, growth, and apoptosis in synovial fluid and numerous tissues in the body [20]. Some studies [20,21] describe an association between Gal-3 and degeneratively altered cartilage. This association is explained by the fact that Gal-3 accumulates in chondrocytes with progressive cartilage degeneration. However, there is still no evidence to clarify the role of Gal-3 in gonarthrosis patients with diabetes mellitus. The aim of this study was to quantify serum levels of Gal-3, pro- and anti-inflammatory cytokines (including their ratios and correlations),and participant’s condition (pain, stiffness, functional limitations) in gonarthrosis patients with and without DM.

## 2. Material and Methods

### 2.1. Procedures

This study was carried out from February to December 2018. The inclusion criteria for patients to participate in the study were: (1) aged 40–70 years; (2) diagnosed gonarthrosis (grades 2 and 3) according to criteria of the American College of Rheumatology [22]; and/or (3) diagnosed DM type 2 according to the next criteria blood glucose ≥ 7.0 mmol/L or plasma glucose in 120 min of OGTT ≥ 11.1 mmol/L; glucose intolerance when blood glucose is <7.0 mmol/L and plasma glucose in 120 min of OGTT ≥ 7.8 mmol/L and ≤11.1 mmol/L; impaired fasting glycemia when plasma fasting glycemia is 6.1 to 6.9 mmol/L and plasma glucose in 120 min of OGTT < 7.8 mmol/L) [23]. The exclusion criteria for patients were: (1) implanted knee endoprosthesis; (2) knee joint injury in the last six months; (3) secondary rheumatoid or septic arthrosis or systemic disease involving the knee joint; (4) active gonarthrosis with intra-articular effusion; (5) treated with antibiotics, aminosalicylates, corticosteroids, immunosuppressants, statins, and biological therapy. After initial screening, blood samples were drawn from the peripheral vein in the morning (8:00–10:00 h) after overnight fasting. The following cytokines were determined from a blood sample: TNF-α, IL-6, IL-12, IL-23, IFN-γ, IL-17, IL-10, and Gal-3.

### 2.2. Participants

After initial screening, 22 patients were excluded from the study due to implanted knee endoprosthesis (*n* = 3), knee joint injury in the last six months (*n* = 7), secondary rheumatoid or septic arthrosis or systemic disease involving the knee joint (*n* = 5), active gonarthrosis with intra-articular effusion (*n* = 1), use of antibiotics, aminosalicylates, corticosteroids, immunosuppressants, statins, and biological therapy (*n* = 6).

A total of 66 patients met the inclusion criteria and were allocated into two groups: gonarthrosis patients without DM (*n* = 23; age: 69.8 ± 8.6 years; BMI: 29.3 ± 4.4) and gonarthrosis patients with DM (*n* = 43; age:68.7 ± 9.4 years; BMI: 29.9 ± 5.3). Before joining the study all patients signed informed consent to participate in the study. The protocol of this study was approved by the Ethics Committee of the University of Kragujevac (No. 01/17-4318) with all procedures conducted in accordance with the Declaration of Helsinki.

### 2.3. Measurement of TNF-α, IL-6, IL-12, IL-23, IFN-γ, IL-17, IL-10, and Gal-3 in Serum Samples

Serum TNF-α, IL-6, IL-12, IL-23, IFN-γ, IL-17, IL-10, and Gal-3 were measured using sensitive enzyme-linked immunosorbent assay (ELISA) kits specific for human cytokines (R&D Systems, Minneapolis, MN, USA) according to the manufacturer’s instructions [24]. As the ratio of counter-regulatory cytokines can be a relevant indicator of disease course [25] we have assessed ratios of pro- and anti-inflammatory cytokines (IL-10/TNF-α, IL-10/IL-6, IL-10/IL-12, IL-10/IL-17).Concentrations of cytokines were determined by interpolation of a standard curve. Results were expressed as picograms per milliliter of patient serum.

### 2.4. WOMAC Questionnaire

The Western Ontario and McMaster Universities Osteoarthrosis index (WOMAC) questionnaire is a three-dimensional disease-specific instrument used in a patient with gonarthrosis or OA of the hip, to measure pain, stiffness, and physical function during typical daily activities [26]. The questions refer to the patient’s condition in the last 48 h. The patient completed the WOMAC questionnaire by selecting one of the offered answers (marking with X). The WOMAC questionnaire consisted of three parts. The first part of the questionnaire included questions about the severity of knee pain (five questions in total), when walking on a flat surface, when climbing and descending stairs, during the night, when changing position (sitting or lying down), and when standing. The second part of the questionnaire included questions related to joint stiffness (two questions). The third part of the questionnaire included questions related to the function of the patient across daily activities (17 questions). The questions referred to difficulties in walking up and down the stairs, getting up from a sitting position, getting into bed, standing, bending down, sitting, resting, the possibility of doing light and heavy work, difficulties getting in and out of a car or bus, when entering the bathtub, when using the toilet, when buying, and when putting on or taking off socks.

### 2.5. Statistical Analysis

Power analysis (G*Power software, version 3.1.7; Heinrich Heine University Düsseldorf, Düsseldorf, Germany) for independent samples *t*-test (using a two-tailed alpha value of 0.05, an effect size of 1.0, and power of 0.80) recommended a sample size of 34 [27], supporting the present analyses (*n* = 66). Software package SPSS version 20 (IBM Corp., Armonk, NY, USA) was used to analyze data. All data were reported as the median and interquartile range. Kolmogorov–Smirnov or Shapiro–Wilk test was used in order to evaluate normality of data distribution. Differences in age, BMI, and WOMAC score between patients with GA and GA and diabetes were assessed using the Student’s *t*-test for normally distributed data and the Mann–Whitney test for all other non-normally distributed data. Spearman’s correlation coefficient estimated the possible relationship between serum levels of Gal-3 and other pro- and anti-inflammatory cytokines. The Spearman correlation coefficient was interpreted as follows: trivial (0.0–0.09), small (0.10–0.29), moderate (0.30–0.49), large (0.50–0.69), very large (0.70–0.89), almost perfect (0.90–0.99), and perfect [28]. Values of *p* < 0.05 were regarded as statistically significant.

## 3. Results

Demographic and clinical characteristics are presented in Table 1. Overall WOMAC index was significantly lower (*p* = 0.001) in gonarthrosis patients without DM compared to gonarthrosis patients with DM.

Median and interquartile ranges for proinflammatory cytokines are presented in Table 2. Significantly lower concentration of TNF-α (*p* = 0.001), IL-6 (*p* = 0.001), IL-12 (*p* = 0.001), IL-17 (*p* = 0.001), IFN-γ (*p* = 0.014), and Gal-3 (*p* = 0.040) was found in gonarthrosis patients with DM compared to gonarthrosis patients without DM.

Median and interquartile ranges for ratios between pro- and anti-inflammatory cytokines in gonarthrosis patients with and without DM are presented in Table 3. IL-10/TNF-α (*p* = 0.001), IL-10/IL-6 (*p* = 0.001), and IL-10/IL-12 (*p* = 0.001) were significantly higher, while IL-10/IL-17 (*p* = 0.001) ratio was significantly lower in gonarthrosis patients with DM compared to gonarthrosis patients without DM.

Median and interquartile ranges for ratios between Gal-3 and pro- and anti-inflammatory cytokines in gonarthrosis patients with and without DM are presented in Table 4. Significantly higher Gal-3/TNF-α (*p* = 0.001), Gal-3/IL-6 (*p* = 0.004), and Gal-3/IL-12 (*p* = 0.001) and lower Gal-3/IL-17 (*p* = 0.001) were detected in gonarthrosis patients with DM compared to gonarthrosis patients without DM.

Further, correlations between Gal-3 and pro- and anti-inflammatory cytokines for all patients in the study are presented in Table 5. Moderate–large positive correlation was detected between the serum values of Gal-3 and IL-12 (*p* = 0.001, r = 0.575), IL-10 (*p* = 0.001, r = 0.535),TNF-α (*p* = 0.046, r = 0.306), and IL-23 (*p* = 0.035, r = 0.323) (Table 5).

Correlations between WOMAC score and cytokines for all patients in the study are presented in Table 6. Moderate inverse correlation was detected between WOMAC score and cytokines, including TNF-a (*p* = 0.001, r = −0.439), IL-6 (*p* = 0.001, r = −0.404), and IL-12 (*p* = 0.012, r = −0.307).

## 4. Discussion

In this study, we quantified serum levels of Gal-3, pro- and anti-inflammatory cytokines (including their ratios and correlations), and participant’s condition (pain, stiffness, functional limitations) in gonarthrosis patients with and without DM. We observed lower serum levels of Gal-3 as well as TNF-α, IL-6, IL-12, IFN-γ, and IL-17 in gonarthrosis patients with DM compared to gonarthrosis patients without DM. Lower proinflammatory cytokines in gonarthrosis patients with DM were accompanied by higher ratios of IL-10/TNF-γ, IL-10/IL-12, and IL-10/IL-6, underpinning the predominance of anti-inflammatory over proinflammatory mediators. Moderate–large correlation of Gal-3 with proinflammatory cytokines showed its potential role in preventing cartilage destruction over other cytokines. In addition, we observed lower pain, stiffness, and functional limitations in gonarthrosis patients without DM compared to gonarthrosis patients with DM.

It is believed that immunopathology is the additional mechanism important in the development and progression of various arthropathies [29,30]. Loria et al. [29] showed that patients with gonarthrosis had bigger inflammatory infiltration and increased production of IL-6 in synovial fluid. On the other hand, a hyperglycemic state in diabetes mellitus suppresses immune defense mechanisms, thus making patients more prone to infections [31]. Different studies confirmed decreased production of proinflammatory cytokines IL-1, IL-2, IL-6, IL-12, and IL-17 in patients with DM, which points to their immunosuppressive state [32,33]. Results from our study clearly showed that gonarthrosis patients with DM had significantly lower systemic values of proinflammatory cytokines TNF-α, IL-6, IL-12, IFN-γ, and IL-17 (Table 2). Additionally, higher ratios of IL-10/TNF-γ, IL-10/IL-12, and IL-10/IL-6 confirmed the predominance of anti-inflammatory over proinflammatory mediators in the serum of gonarthrosis patients with DM (Table 3). These results are in line with previous studies [32,33], suggesting that gonarthrosis patients with DM have a predominance of immunosuppressive state. However, our result showed a higher WOMAC score and a more severe form of gonarthrosis in patients with DM (Table 1). We do believe that these systemic levels of cytokines do not reflect in the right way the local inflammation that dominates in the knees of these patients. Further, there is ample evidence suggesting the important role of some specific cytokines in ankle tissue remodeling and regeneration. Some studies claim that higher levels of proinflammatory cytokines are associated with knee cartilage loss [34,35]. However, during the time, studies revealed the cartilage-protective effect of some cytokines. Among them, Relic et al. [36] showed that TNF-α can neutralize the apoptotic effect of nitrite oxide and thus prevent the death of chondrocytes. Another study revealed that IL-6 dominates in the synovial fluid during cartilage repair and that injection of exogenous IL-6 stimulates production of glycosaminoglycans by healthy chondrocytes [37]. Page et al. [38] demonstrated that elimination of IFN-γ is associated with a higher level of bone and cartilage destruction in mice with rheumatoid arthritis, while an in vitro study showed that IFN-γ inhibits synthesis of matrix metalloproteinases and prevents cartilage damage. In our study, lower values of TNF-α, IL-6, and IFN-γ in patients with gonarthrosis and DM may explain the lower values of the WOMAC index.

Apart from diabetes-induced immunosuppression, it has been established that patients with DM also suffer from oxidative and metabolic stresses. This means that production of free oxidative species stimulates specific biochemical pathways such as glycation, nitrosylation, carbonylation, and lipoxidation [39]. These processes interfere with normal synthesis of molecules that are part of cartilage tissue, thus increasing susceptibility for cartilage destruction [40,41]. Taking this into account, we believe that through oxidative stress, diabetic condition impedes synthesis of cartilage, while the diabetes-induced immunosuppressive state does not facilitate tissue damage but inhibits tissue remodeling and healing. DM is an obvious cause of systemic immunosuppression and lower values of inflammatory mediators in patients with gonarthrosis. We assume that diabetes did not dominantly affect tissue damage through reduced cytokine values but did affect the rate of tissue regeneration.

Gal-3 concentration is significantly decreased in gonarthrosis patients with DM compared to gonarthrosis patients without DM (Table 2). Previous studies showed that Gal-3 levels were higher in patients with DM and prediabetes compared to control groups [42,43]. Opposing results in Gal-3 levels in patients with DM compared to our study may be due to gonarthrosis comorbidity. A lot of studies have demonstrated pleiotropic biological functions of Gal-3. Our previous studies about Gal-3 revealed the hepatoprotective function of Gal-3 from hepatitis C virus damage, as well as the immunosuppressive role of Gal-3 on antitumor immunity [44,45]. Filer et al. [46] showed that Gal-3 stimulates persistence of the inflammatory infiltrate in patients with rheumatoid arthritis. On the other hand, a recent study explained that by binding to a specific mucinous glycopeptide named lubricin, Gal-3 provides cartilage lubrication and postpones development and progression of arthritis [47]. mice without Gal-3 developed cartilage lesions thus confirming the protective role of Gal-3 in chondrocyte survival [48]. As our study showed a lower level of Gal-3 as well as TNF-α, IL-6, IL-12, IFN-γ, and IL-17 in gonarthrosis patients with DM, we believe that the lack of these cytokines caused intense tissue damage and slower tissue repair, which is reflected through a higher WOMAC score (Table 2). This assumption is confirmed by the strong positive correlation that is measured between Gal-3 and IL-12 and the moderate correlation between Gal-3 and TNF-α and IL-23, which indicates their synergistic effect (Table 5). Moreover, significantly increased ratios of Gal-3/TNF-α, Gal-3/IL-12, and Gal-3/IL-6 showed predominance of Gal-3 over proinflammatory cytokines, which points to the potentially more important role of Gal-3 in preventing cartilage destruction (Table 4). Previous studies have already analyzed the interconnection of Gal-3 and proinflammatory cytokines. Simovic et al. [49] showed that elimination of Gal-3 resulted in significantly lower numbers of IL-12-producing macrophages, lower numbers of dendritic cells (one of the major sources of TNF-α, IL-12, and IL-23), and a lower number of TNF-α producing neutrophils in the colon of mice. Arad et al. [50] showed that inhibition of Gal-3 decreased Toll-like receptor-2, -3, and -4-related IL-6 secretion, thus suggesting that Gal-3 is a stimulator of TLR-induced proinflammatory cytokine secretion in human synovial fibroblasts. By pointing out that Gal-3 directly affects the cells that are sources of proinflammatory cytokines and modulates their production, these studies are in line with our results and confirm the positive relationship of Gal-3/IL-12 and TNF-α/IL-23.

The significantly higher WOMAC index in gonarthrosis patients with DM compared to gonarthrosis patients without DM we observed is in line with previous studies [51], showing a higher WOMAC index in gonarthrosis patients than the control population. In this regard, Davies-Tuck et al. [52] showed that cartilage damage was positively associated with higher serum glucose levels in females. Further, Eymardet al. [3] and Veronese et al. [53] showed that DM worsens the clinical picture in patients with gonarthrosis, which coincides with the results of our study. Other studies confirmed that the deterioration of the cartilage matrix in the knee of DM is faster, thus accelerating gonarthrosis [54].

## 5. Limitations

This study also has some limitations. First, the study is observational and was conducted on a relatively small number of patients. Another limitation is the disproportion between the sexes in both groups, with the majority being women. Biopsy of the cartilage has not been performed, and the cytokines were not measured in the synovial fluid.Additionally, there is no correction for confounders. Future research is encouraged to determine clinically relevant differences inthe cytokines in gonarthrosis patients with and without DM.

## 6. Conclusions

In summing up, our data show that gonarthrosis patients with DM have significantly higher WOMAC index values and therefore a more severe form of gonarthrosis. Systemic values of proinflammatory TNF-α, IL-6, IL-12, IFN-γ, IL-17, and Gal-3 in gonarthrosis patients with DM were significantly lower than in gonarthrosis patients without DM. The precise mechanism of Gal-3 effect in gonarthrosis and DM comorbidity is still to be clarified.

## Figures and Tables

**Table 1 ijerph-19-11480-t001:** Demographic and clinical characteristics (mean ± standard deviation (SD)) ingonarthrosis patients with (*n* = 43, females: *n* = 33, males: *n* = 10) and without diabetes mellitus (*n* = 23, females: *n* = 20; males: *n* = 3).

Demographic Characteristics	Total	Gonarthrosis Patients	Gonarthrosis Patients with Diabetes Mellitus	Mean Differences with 95% Confidence Interval	*p* Value
Age (years)	69.10 (9.06)	69.83 (8.59)	68.72 (9.37)	−1.105 (−5.507–3.596)	0.640
Body mass index (kg/m^2^)	29.69 (4.97)	29.28 (4.38)	29.91 (5.3)	0.624 (−1.96–3.21)	0.631
WOMAC scores	52.47 (11.51)	44.57 (11.52)	56.7 (9.11)	12.123 (6.969–17.296)	0.001
Pain	11.14 (2.50)	9.26 (2.3)	12.14 (2)	2.88 (1.79–3.97)	0.001
Stiffness	3.05 (1.31)	2.57 (1.47)	3.3 (1.15)	0.74 (0.08–1.39)	0.026
Functional limitations	38.24 (8.66)	32.65 (8.79)	41.23 (7.02)	8.58 (4.62–12.54)	0.001

**Table 2 ijerph-19-11480-t002:** TNF-α, IL-6, IL-12, IL-117, IFN-γ, IL-23, IL-10, and Gal-3 (median and interquartile range) in gonarthrosis patients with and without diabetes melitus.

Cytokine	Gonarthrosis Patients (*n* = 23)	Gonarthrosis Patients with Diabetes Mellitus (*n* = 43)	Significance of Difference
TNF-α (pg/mL)	52.23 (38.54–86.65)	16.08 (10.10–23.02)	*p* = 0.001
IL-6 (pg/mL)	91.11 (87.31–98.82)	49.89 (42.94–58.11)	*p* = 0.001
IL-12 (pg/mL)	80.47 (62.25–98.39)	28.98 (21.30–46.75)	*p* = 0.001
IL-17 (pg/mL)	5.37 (3.02–9.86)	16.08 (11.83–21.90)	*p* = 0.001
IFN-γ (pg/mL)	77.54 (59.23–106.00)	66.77 (55.06–77.62)	*p* = 0.014
IL-23 (pg/mL)	143.96 (87.71–264.13)	118.34 (93.89–168.70)	*p* = 0.278
IL-10 (pg/mL)	188.66 (118.22–282.03)	184.75 (138.91–323.40)	*p* = 0.531
Gal-3 (pg/mL)	1518.99 (1136.09–1948.07)	1304.78 (805.41–1758.63)	*p* = 0.040

Note: Data analyzed using Mann–Whitney test.

**Table 3 ijerph-19-11480-t003:** IL-10/TNF-γ, IL-10/IL-6, IL-10/IL-12, and IL-10/IL-17 (median and interquartile range) in gonarthrosis patients with and without diabetes melitus.

Ratios between Anti- and Proinflammatory Cytokines	Gonarthrosis Patients (*n* = 23)	Gonarthrosis Patients with Diabetes Mellitus (*n* = 43)	Significance of Difference
IL-10/TNF-α	3.78 (1.90–7.11)	12.89 (6.67–20.34)	*p* = 0.001
IL-10/IL-6	2.03 (1.30–3.12)	3.76 (2.88–5.70)	*p* = 0.001
IL-10/IL-12	2.53 (1.50–3.63)	5.36 (3.49–10.10)	*p* = 0.001
IL-10/IL-17	33.28 (20.93–46.94)	10.93 (7.72–19.82)	*p* = 0.001

Note: Data analyzed using Mann–Whitney test.

**Table 4 ijerph-19-11480-t004:** Gal-3/TNF-α, Gal-3/IL-6 and Gal-3/IL-12 and Gal-3/IL-17 (median and interquartile range) in gonarthrosis patients with and without diabetes melitus.

Ratios betweenGal-3 and Cytokines	Gonarthrosis Patients (*n* = 23)	Gonarthrosis Patients with Diabetes Mellitus (*n* = 43)	Significance of Difference
Gal-3/TNF-α	24.26 (18.10–39.73)	67.18 (37.78–131.84)	*p* = 0.001
Gal-3/IL-6	15.89 (11.15–20.78)	24.40 (16.15–35.38)	*p* = 0.004
Gal-3/IL-12	19.22 (14.56–25.72)	38.23 (24.24–53.55)	*p* = 0.001
Gal-3/IL-17	260.21 (150.54–500.48)	68.60 (41.12–109.99)	*p* = 0.001

Note: Data analyzed using Mann–Whitney test.

**Table 5 ijerph-19-11480-t005:** Correlation between Gal-3 and pro- and anti-inflammatory cytokines.

Cytokines	Spearman r	*p* Value
TNF-α	0.306	0.046
IL-6	0.069	0.662
IL-12	0.575	0.001
IL-17	0.094	0.548
IFN-γ	0.193	0.214
IL-23	0.323	0.035
IL-10	0.535	0. 001

**Table 6 ijerph-19-11480-t006:** Correlations between WOMAC scores and cytokines.

WOMAC Score			WOMAC Score Pain		WOMAC Score Stiffness		WOMAC Score Functional Limitations	
	Spearman’s Rho	*p* Value	Spearman’s Rho	*p* Value	Spearman’s Rho	*p* Value	Spearman’s Rho	*p* Value
TNF-α	−0.439	0.001	−0.542	0.001	−0.235	0.057	−0.411	0.001
IL-6	−0.404	0.001	−0.459	0.001	−0.188	0.130	−0.386	0.001
IL-12	−0.307	0.012	−0.400	0.001	−0.132	0.291	−0.287	0.019
IL-17	0.224	0.070	0.230	0.064	0.142	0.254	0.236	0.056
IFN-γ	−0.070	0.578	−0.188	0.130	0.084	0.504	−0.063	0.617
IL-23	−0.158	0.206	−0.141	0.258	0.179	0.151	0.173	0.166
IL-10	−0.044	0.726	−0.020	0.874	−0.109	0.384	−0.031	0.805
Gal-3	−0.054	0.666	−0.140	0.261	−0.040	0.748	−0.027	0.829

## Data Availability

The data sets generated during the study are available from the corresponding author upon reasonable request.

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
