# Peer review of "Potential Protective Role of Galectin-3 in Patients with Gonarthrosis and Diabetes Mellitus: A Cross-Sectional Study"

_ijerph, 2022, doi:10.3390/ijerph191811480_

Round 1

Reviewer 1 Report

Comments to the manuscript ID:  ijerph-1878091 entitled: Potential protective role of galectin-3 in patients with gonarthrosis and diabetes mellitus. This is an interesting manuscript about the influence of the galectin -3 in patients with gonarthrosis with and without diabetes mellitus. The manuscript is well written and structure. Minor changes are required.

Title:

Please add the kind of study development: A cross over study?

Abstract:

Please erase the acronymus as for example (DM)

Keywords:

Please correct the keyword in black and add more

Introduction:

In the first line: gonarthrosis is only a rheumatic disease? Can other diseases produce gonarthrosis?

Material and methods:

Can author explain how calculate the sample size? This in an important fact due to the limitations.

In statistical analysis section, please add the version 20 of the SPSS software out of the parenthesis.

Results:

Please add the sex in the table 1, and correct the lines.

Please add a new table with the statistical results of the figures. It can understand better the results. With the figures are not clear the results.

Discussion:

Congratulations to the authors, is a great discussion about the research.

Please add the limitation section inside the discussion. ( is important for authors add the sample size calculations based on the limitations).

Conclusions:

Ok

References:

Ok.

Author Response

Response to Reviewer

This is an interesting manuscript about the influence of the galectin -3 in patients with gonarthrosis with and without diabetes mellitus. The manuscript is well written and structure. Minor changes are required.

Our response: We appreciate the positive feedback regarding our manuscriptin general.

TITLE

Please add the kind of study development: A cross over study?

Our response:The point you made regarding study design is valid. In alignment with your comment we added in the title“ A cross sectional study” In our study a cross-sectional design was adopted, since our patients did not cross-over from one treatment to another during the study period.

ABSTRACT

Please erase the acronymus as for example (DM)

Our response:The acronymus DM has been deleted from the Abstract.

KEYWORDS

Please correct the keyword in black and add more

Our response:The suggested changes have been made.

INTRODUCTION

In the first line: gonarthrosis is only a rheumatic disease? Can other diseases produce gonarthrosis?

Our response:Gonarthrosis is a rheumatic disease. There are numerous risk factors that can contribute to the aggravation of gonarthrosis, but not other diseases that cause gonarthrosis.

MATERIAL AND METHODS

Can author explain how calculate the sample size? This in an important fact due to the limitations.

Our response:Thank you for this insight. The following sentence has been added to the revised manuscript:An a priori analysis using G*Power software (version 3.1.7; Heinrich Heine University Düsseldorf, Düsseldorf, Germany) for independets samples t-test (using a 2-tailed alpha value of 0.05, an effect size of 1.0, and power of 0.80) recommended a sample size of 34, supporting the present analyses (n = 66).

In statistical analysis section, please add the version 20 of the SPSS software out of the parenthesis.

Our response:the “version 20” after SPSS softwer of the parenthesis is added in statistical analysis section.

RESULTS

Please add the sex in the table 1, and correct the lines.

Our response:Thank you for this comment.Sample size across sexes is already presented in the title of Table 1.

Please add a new table with the statistical results of the figures. It can understand better the results. With the figures are not clear the results.

Our response:We thank the reviewer for this insight. New table with the statistical results has been added to the revised manuscript.

DISCUSSION

Congratulations to the authors, is a great discussion about the research.

Our response:We thank the reviewer for this comment.

Please add the limitation section inside the discussion. ( is important for authors add the sample size calculations based on the limitations).

Our response: According to the suggestion of the Reviewer, limitation section has been modified. This section is now expanded and explained more precisely. 

Conclusions:

Ok

Our response:We thank the reviewer for this comment.

References:

Ok.

Our response:We thank the reviewer for this comment.

Reviewer 2 Report

The paper is suitable for publication without further changes, as it includes a well-planned statistical analysis that leads to the presented conclusions. I believe this study is of potential interest to the readers of the International Journal of Environmental Research and Public Health, as the first study of this type. The authors have also discussed some limitations of this study.

Author Response

Response to Reviewer 

The paper is suitable for publication without further changes, as it includes a well-planned statistical analysis that leads to the presented conclusions. I believe this study is of potential interest to the readers of the International Journal of Environmental Research and Public Health, as the first study of this type. The authors have also discussed some limitations of this study.

Our response: Thank you for this encouraging comment regarding our manuscript.

Reviewer 3 Report

The authors have compared 23 patients with gonarthrosis to 43 with gonartrosis and DM on serum inflammatory markers and have found many differences between the two groups.

Please follow the STROBE criteria for reporting as this is an observational study.

It is unclear how these 66 patients were selected. How many patients were eligible and what were the reasons they were not included. A flow chart would be helpful here.

What were the criteria for DM? How was this diagnosed / determined?

Why were only serum cytokines determined? It would be interesting to also know the cytokines in the synovial fluid and compare these to the serum values between the DM – and DM + groups.

Please provide mean differences with 95% confidence intervals instead of p-values. See this nature paper:

Nature. 2019 Mar;567(7748):305-307.

doi: 10.1038/d41586-019-00857-9.

Scientists rise up against statistical significance

Valentin Amrhein, Sander Greenland, Blake McShane

Regarding the cytokines, what are (minimally) clinically relevant differences?

In other words, are the differences found clinically relevant as well as statistically significant?

Please provide the ahlback classification of OA in both groups. How much joint destruction was there?

Was there a correlation between cytokines and WOMAC scores?

Was there a correlation between Ahlback classification and cytokines?

Was there a correlation between WOMAC scores and Ahlback classification?

The paper can benefit from a more elaborate limitation section. E.g. there is no correction for confounders and the study is observational.

It is not clear what is meant by the last sentence of the limitations. Please rephrase.

Regarding: “Gal-3 270
strongly correlates with and dominates over proinflammatory cytokines thus showing its 271
potential role in preventing cartilage destruction over other cytokines.

The authors are not measured cartilage destruction. Hence, this conclusion is not in line with the results and should thus be removed.

Author Response

Response to Reviewer

The authors have compared 23 patients with gonarthrosis to 43 with gonartrosis and DM on serum inflammatory markers and have found many differences between the two groups.

Please follow the STROBE criteria for reporting as this is an observational study.

            Our response: We appreciate you raising this important point.We have checked our approach aligns STROBE criteria.

It is unclear how these 66 patients were selected. How many patients were eligible and what were the reasons they were not included. A flow chart would be helpful here.

            Our response: Thank you for this insight. The following sentence has been added to the revised manuscript:An a priori analysis using G*Power software (version 3.1.7; Heinrich Heine University Düsseldorf, Düsseldorf, Germany) for independets samples t-test (using a 2-tailed alpha value of 0.05, an effect size of 1.0, and power of 0.80) recommended a sample size of 34, supporting the present analyses (n = 66).

In addition, we feel our section detailing the inclusion and excusion criteria address this point. The inclusion criteria for patients to participate in the study were: 1) aged 40-70 years; 2) diagnosed gonarthrosis (grade 2 and 3) according to criteria of American College of Rheumatology [22]; and/or 3) diagnosed DM type 2 according to the criteria of ADA-EASD [23]. The exclusion criteria for patients were:  1) implanted knee endoprosthesis; 2) knee joint injury in the last six months; 3) secondary rheumatoid or septic arthrosis or systemic disease involving the knee joint; 4) active gonarthrosis with intra-articular effusion; 5) treated with antibiotics, aminosalicylates, corticosteroids, immunosuppressants, statins and biological therapy.

What were the criteria for DM? How was this diagnosed / determined?

 Our response: DM was diagnosed according to the criteria of ADA-EASD.

The diagnostic criteria were:

  • blood glucose ≥ 7.0 mmol/l or plasma glucose in 120 min of OGTT ≥ 11.1 mmol/l
  • glucose intolerance when blood glucose is < 7.0 mmol/l and plasma glucose in 120 min of OGTT ≥ 7.8 mmol/l and ≤11.1 mmol/l
  • impaired fasting glycemia when plasma fasting glycemia is 6.1 to 6.9 mmol/l and plasma glucose in 120 min of OGTT < 7.8 mmol/l

Why were only serum cytokines determined? It would be interesting to also know the cytokines in the synovial fluid and compare these to the serum values between the DM – and DM + groups.

 Our response: We agree with the usefulness of including the values cytokines in the synovial fluid, but,unfortunately, we did not technically have the possibility of taking synovial fluid from patients.

Please provide mean differences with 95% confidence intervals instead of p-values. See this nature paper:

Nature. 2019 Mar;567(7748):305-307.doi: 10.1038/d41586-019-00857-9.Scientists rise up against statistical significanceValentin Amrhein, Sander Greenland, Blake McShane

  • Our response: Thank you for the thought-provoking suggestions and for recommending the highly relevant, current, and useful work byAmrhein.Given null hypothesis significance testing (p values) provides only part of the complete picture, our data interpretation and conclusions were formulated using median and interquartile range. Since the Mann-Whitney test was used to compare the mean ranks, we reportedthe median and interquartile range without mean difference.

Regarding the cytokines, what are (minimally) clinically relevant differences?

In other words, are the differences found clinically relevant as well as statistically significant?

  • Our response: We value your suggestion on this matter and believe you are proposing use of effect size.However, our data were not normally distributed to add effect size (e.g. Cohen's d) andformulate conclusions using the practical statistics. We are fully aware of the fact that statistical significance does not necessarily imply clinical importance. Minimal clinically important difference represents the smallest change in an outcome that a patient would identify as important. We believe that we cannot consider cytokine values as outcome variables, given that cytokines themselves directly and indirectly affect outcome variables that are clinically measurable. Based on only one our study, it is difficult to conclude whether alterations in the systemic values of cytokines are really clinically relevant, due to their mutual intercorrelation in the cytokine milieu as well as due to the influence on other relevant factors involved in the genesis and progression of the disease.

Please provide the ahlback classification of OA in both groups. How much joint destruction was there?

  • Our response: Thank you for this insight and for recommending the highly relevant Ahlback classification of OA. However, the Kellgren-Lawrence's OA classification adopted in our study is valid (1) and also widelyutilized across previous studies(2-6). In addition, in our case Ahlback classification could not be implemented sincedata concerning OA gradeswere retrieved from patient'smedical history forms, whereby the Kellgren-Lawrence OA classification has been standarly used in determining OA stage at our clinic.
  1. Kohn, M.D., Sassoon, A.A. & Fernando, N.D. Classifications in Brief: Kellgren-Lawrence Classification of Osteoarthritis. Clin Orthop Relat Res 474, 1886–1893 (2016). https://doi.org/10.1007/s11999-016-4732-4
  2. Al-Jarallah K,Shehab D,Abdella N,Mohamedy HA,Abraham Knee Osteoarthritis in Type 2 Diabetes Mellitus: Does Insulin Therapy Retard Osteophyte Formation? Med Princ Pract. 2015; 25(1): 12–17.
  3. Abdalbary Ultrasound with mineral water or aqua gel to reduce pain and improve the WOMAC of knee osteoarthritis. Future Sci OA. 2016; 2(1): FSO110.
  4. Rooij MD, Leeden MVD, Cheung J, Esch MVD, Häkkinen A, Haverkamp D, Roorda LD, Twisk J, Vollebregt J, Lems WF, Dekker Efficacy of Tailored Exercise Therapy on Physical Functioning in Patients With Knee Osteoarthritis and Comorbidity: A Randomized Controlled Trial. Arthristis Care & Research. 2017; 69(6): 807-816.
  5. Al-Jarallah K, Shehab D, Abdella N, Al Mohamedy H, Abraham M. Knee Osteoarthritis in Type 2 Diabetes Mellitus: Does Insulin Therapy Retard Osteophyte Formation? Med Princ Pract 2016; 25:12–17.
  6. Kohn MD,Sassoon AA, Fernando ND. Classifications in Brief: Kellgren-Lawrence Classification of Osteoarthritis. Clin Orthop Relat Res. 2016 Aug; 474(8): 1886–1893.

Was there a correlation between cytokines and WOMAC scores?

  • Our response: We value the insight offered by the reviewer here.However, to avoid replication in reporting the inverse correlation (e.g. between TNF-a and WOMAC score) or lower pro-inflammatory cytkines and higher WOMAC score (in GA+DM or vice versa in GA patients) these data are not included to the revised manuscript.

WOMAC score

WOMAC score

 pain

WOMAC score stiffness

WOMAC score functional limitations

Spearman's rho

p value

Spearman's rho

p value

Spearman's rho

p value

Spearman's rho

p value

TNF-α

-0.439

0.001

-0.542

0.001

-0.235

0.057

-0.411

0.001

IL-6

-0.404

0.001

-0.459

0.001

-0.188

0.130

-0.386

0.001

IL-12

-0.307

0.012

-0.400

0.001

-0.132

0.291

-0.287

0.019

IL-17

0.224

0.070

0.230

0.064

0.142

0.254

0.236

0.056

IFN-γ

-0.070

0.578

-0.188

0.130

0.084

0.504

-0.063

0.617

IL-23

-0.158

0.206

-0.141

0.258

0.179

0.151

0.173

0.166

IL-10

-0.044

0.726

-0.020

0.874

-0.109

0.384

-0.031

0.805

Gal-3

-0.054

0.666

-0.140

0.261

-0.040

0.748

-0.027

0.829

Was there a correlation between Ahlback classification and cytokines?Was there a correlation between WOMAC scores and Ahlback classification?

  • Our response: As outlined in previous response, we adopted Kellgren-Lawrence classificationas the most widely used clinical tool for the radiographic diagnosis of OA.In addition, we recruited only patients with stage 2 and 3, while patients with grade 0, 1 and 4 were not inclluded, which limit our ability to make solid conclusionsconcerning the correlation between cytokines and OA stage.

The paper can benefit from a more elaborate limitation section. E.g. there is no correction for confounders and the study is observational.

  • Our response: According to the suggestion of the Reviewer, limitation section has been modified. This section is now expanded and explained more precisely.

It is not clear what is meant by the last sentence of the limitations. Please rephrase.

  • Our response: In line with the Reviewer's comment, the last sentence in the limitation section is now rephrased. 

Regarding:“Gal-3 270strongly correlates with and dominates over proinflammatory cytokines thusshowingits 271potential role in preventing cartilage destruction over other cytokines.”

The authors are not measured cartilage destruction. Hence, this conclusion is not in line with the results and should thus be removed.

  • Our response: According to the suggestion of the Reviewer, the sentence "Gal-3 strongly correlates with and dominates over proinflammatory cytokines thus showing its potential role in preventing cartilage destruction over other cytokines" is deleted from the conclusions.

Round 2

Reviewer 3 Report

The authors have addressed some of my comments and the manuscript has improved. However, there are some issues remaining that need to be addressed.

The sample size calculation seems to be post-hoc. There is no proof provided (e.g. protocol) that this was done a priori.

The sample size calculation does not provide an answer to my question regarding the selection of the 66 patients and neither does mentioning of the inclusion and exclusion criteria. How many patients were eligible and what were the reasons they were not included. A flow chart would be helpful here.

Please add the DM diagnostic criteria to the text.

Apparently it is unknow what are clinically relevant differences for the cytokines. This is a limitation of the study as we cannot know if the results are meaningful or not.

Please include the table with the correlations between WOMAC scores and cytokines in the paper. There is not (valid) reason to leave it out.  

Author Response

The authors have addressed some of my comments and the manuscript has improved. However, there are some issues remaining that need to be addressed.

Our response: Thank you for the  valuable insight and thoughtful comments

The sample size calculation seems to be post-hoc. There is no proof provided (e.g. protocol) that this was done a priori.

 Our response: Thank you for this comment.This sentence has been modified as follows: "Power analysis (G*Power software, version 3.1.7; Heinrich Heine University Düsseldorf, Düsseldorf, Germany) for independets samples t-test (using a 2-tailed alpha value of 0.05, an effect size of 1.0, and power of 0.80) recommended a sample size of 34 (27), supporting the present analyses (n = 66)."

The sample size calculation does not provide an answer to my question regarding the selection of the 66 patients and neither does mentioning of the inclusion and exclusion criteria. How many patients were eligible and what were the reasons they were not included. A flow chart would be helpful here.

 Our response: In alignment with your comment, the following sentence has been added to the revised manuscript:"After initial screening, 22 patients were excluded from the study due to implanted knee endoprosthesis (n = 3), knee joint injury in the last six months (n = 7), sec-ondary rheumatoid or septic arthrosis or systemic disease involving the knee joint (n = 5), active gonarthrosis with intra-articular effusion (n = 1), use of antibiotics, aminosa-licylates, corticosteroids, immunosuppressants, statins and biological therapy (n = 6).

Please add the DM diagnostic criteria to the text.

 Our response: Thank you for the  suggestion DM diagnostic criteria has been added to the revised manuscript

Apparently it is unknow what are clinically relevant differences for the cytokines. This is a limitation of the study as we cannot know if the results are meaningful or not.

            Our response: The following sentence has been added to the limitation section: "Future research is encouraged determining clinically relevant differences for the cytokines in gonarthrosis patients with and without DM."

Please include the table with the correlations between WOMAC scores and cytokines in the paper. There is not (valid) reason to leave it out.

Our response: Thank you for the  suggestion.Table with the correlations between WOMAC scores and cytokines has been added to the revised manuscript. Data have been interpreted as follows: "Correlations between WOMAC socre  and cytokines for all patients in the study are presented in Table 6. Moderate inverse correlation was detected between WOMAC score and cytokines, including TNF-a (p = 0.001, r = -0.439), IL-6 (p = 0.001, r = -0.404), and IL-12 (p = 0.012, r = -0.307)."